# Skew Rolling of Bimetallic Rods

**DOI:** 10.3390/ma14010018

**Published:** 2020-12-22

**Authors:** Janusz Tomczak, Tomasz Bulzak, Zbigniew Pater, Łukasz Wójcik, Tomasz Kusiak

**Affiliations:** Faculty of Mechanical Engineering, Lublin University of Technology, 36 Nadbystrzycka Str., 20-618 Lublin, Poland; t.bulzak@pollub.pl (T.B.); z.pater@pollub.pl (Z.P.); l.wojcik@pollub.pl (Ł.W.); t.kusiak@pollub.pl (T.K.)

**Keywords:** skew rolling, bimetallic rods, welding, FEM

## Abstract

The present article reports selected results of a preliminary study of the process of skew rolling of bimetallic rods. The experiments were conducted using a numerically controlled three-roller skew rolling mill. During the tests, bimetallic rods were rolled from billets whose cores and outer sleeves (bushings) were made of different types of steel. The results demonstrate that the proposed method can be successfully used in the production of bimetallic rods. However, proper fastening of the two materials depends on the geometrical parameters of the billets, and the quality of bimetallic rods depends on the heating method used. When the rods are heated without protective atmospheres, the surface layer of the core gets decarburized and the surfaces of the materials being joined together are oxidized, which hinders the welding process and adversely affects the physical and chemical properties of such products. The results of numerical modeling indicate that the material near the surface tends to flow, which may have a negative impact on the welding process. In addition, the distribution of stress in the tool–workpiece contact zone may make welding of the materials difficult. The results reported in this paper are preliminary and constitute a prelude to a more detailed analysis of bimetallic rod rolling.

## 1. Introduction

Recently, bimetallic materials, i.e., materials produced by permanently fastening two or more metals or metal alloys, have been increasingly used in machine construction. This is because components made of such materials have unique physical and chemical properties [1]. However, for bimetallic parts to operate properly, they have to be made of high quality materials that can be permanently joined over the entire contact surface. Depending on the intended use and required mechanical properties, bimetallic materials can be joined by bonding, adhesive bonding, press-fitting, or shrink-fitting [2,3,4,5,6]. Bimetallic materials for the production of machine parts that carry larger loads are typically welded or fusion-welded [7,8,9,10,11]. Joining technologies can be used to produce a permanent joint between two metals or alloys (usually of a similar grade) which is characterized by high strength parameters. One example of the application of such materials is the production of parts used in aviation. Bimetallic aircraft parts are made of very light magnesium alloys clad with a layer of aluminum alloys [12,13]. Because magnesium alloys have very low corrosion resistance, the surface layer of such parts is mechanically covered with a layer of aluminum alloys, which are much more resistant to corrosion. As a result, a low weight part with increased corrosion resistance is produced [14]. Another area of application of bimetallic materials is the manufacture of various types of cutting tools in which the working part is made of tool materials characterized by very high strength properties, and the gripping part is made of much cheaper general-use construction materials [15,16,17]. The automotive industry also uses bimetallic materials for the production of drive system components. One example are axle shafts, in which the hub is fastened to the axle shaft in the process of frictional butt-welding [18,19]. Studies have also been carried out on plastic forming of two materials during which they are welded to yield a monolithic part characterized by unique physico-chemical properties [20,21,22,23]. It should be noted, however, that in many cases, the process of joining two different materials is relatively expensive and complicated [24,25], and does not guarantee the formation of a permanent joint. An additional drawback is the relatively small size of the bimetallic blanks obtained using these technologies [26,27,28]. This limits the applicability of bimetallic materials to the production of small parts. Searching for an effective method of producing bimetallic blanks (which are most commonly obtained by joining two alloys with different properties), skew rolling experiments were conducted in which workpieces were passed between three rollers in a CNC (Computerized Numerical Control) rolling mill. The use of CNC rolling mills does not limit the technological capability of the analyzed process for producing semi-finished products with a constant cross-section. It seems that with this type of machine it is possible to produce bimetallic forgings of stepped axles and shafts, as well as axisymmetric forgings that could be used as billets in die forging processes or other forge rolling processes. The present study is a pilot to broader research on the production and forming of bimetallic parts using rotation and forging technologies.

## 2. Scope of the Study

Preliminary tests were conducted in which bimetallic rods were formed in a skew rolling process. Five variants were analyzed in which workpieces obtained by joining two different grades of steel were rolled. The first four variants of the process related to rolling bimetallic bars in which the outside tube had an outside diameter of 42.4 mm and was made of the alloy steel grade X5CrNi18–10 (Centravis Production Ukraine, Dnipropetrovsk, Ukraine), while the core was made of C45 carbon steel (ArcelorMittal International Ukraine, Kiev, Ukraine). For these pair of materials, the rolling process was conducted with two different values of the reduction ratio *δ* = *D/d*, i.e., *δ* = 1.146 (process variants I and II) and *δ* = 1.285 (process variants III and IV). For the two reduction ratios, two different values of the outside tube wall thickness t were taken, i.e., *t* = 3.7 mm (process variants I and III) and *t* = 2.2 mm (process variants II and IV). The core and the tube were joined with H7/p6 fit. Process variant V involved rolling bimetallic bars in which the outside tube had an outside diameter of 51 mm and was made of the unalloyed steel grade S355, while the core was made of the medium-alloy steel grade 42CrMo4. In this process variant, bars were formed with the reduction ratio of *δ* = 1.275 and the initial tube wall thickness of *t* = 8 mm. Like previously, the core and the tube were joined with H7/p6 fit. In this way, it was possible to preliminarily determine the effect of material grade, reduction ratio, and outer tube wall thickness on the skew rolling process and the weldability of both materials (in terms of producing bimetallic bars). The parameters of the workpieces and the rods rolled from them are given in Table 1. The workpieces were rolled using three tools positioned obliquely to the rolling axis (the tool chamfer angle *α* was 5°). A schematic of the process of skew rolling of bimetallic rods is shown in Figure 1.

The rolling tools were conical cylinders, in which the conical surfaces were the working parts (compressing the material), and the cylindrical surface was only used to calibrate the already formed part of the workpiece. The calibrating surfaces of the rollers had a diameter of 150 mm, and the conical compressing surfaces had an inclination angle of 20°. During rolling, the billet (a section of a pipe enveloping a steel rod) (Figure 2) was first heated in an electric chamber furnace and then secured in a holder (a gripper) which moved along the rolling axis, providing controlled movement of the materials being rolled. The tools rotated in the same direction at a speed of 60 rpm, and the gripper moved at a speed of 15 mm/s. The kinematic parameters of the process were adopted in such a way that the value of the vector of the axial velocity induced at the tool-workpiece contact (due to twisting the rolls at the angle *α*), being one of the components of the tool circumferential velocity vector, was 5% lower than the axial velocity of the gripper. As a result, throughout the process, the gripper moved the material in a controlled manner, generating tensile stress in the rolled section. The process parameters adopted in the tests allowed to determine the influence of the geometric parameters of the workpiece (diameter and wall thickness), material parameters (similar/different steel grades), and rolling parameters (the value of work ratio) on the quality of the obtained bimetallic rods.

## 3. Experimental Tests of Skew Rolling Process of Bimetallic Rods

The rolling tests were performed using a numerically controlled three-roller skew rolling mill (Figure 3) (WSN-10, Lublin University of Technology, Lublin, Poland), available at the Department of Computer Modeling and Metal Forming Technologies of Lublin University of Technology. The skew rolling mill is a highly versatile machine, which can be used to roll products with different contours employing the same set of tools. It consists of eight basic units: a support frame 1, a drive unit 2, a rolling cage 3, an axial travel unit 4, gripper and workpiece guide units 5, a torque transmission 6, a material feeding unit 7, and a workpiece support unit 8. The machine offers wide technological possibilities. Because it allows to numerically control tool and workpiece movement, the rolling mill can be used to form axisymmetric forgings with any external contour.

An interference fit was assumed for initial joining of the tube with the core. The fit value of H7/p6 was determined based on the recommended fits in machine design. At a higher value of interference, it was difficult to join both elements (buckling occurred). Before press fitting, both the core and tube were machined (the surface of the cores was ground to obtain the exact dimension with a p6 tolerance field, while the holes in the tubes were bored with H7 tolerance). During the tests, degreased and press-fit billets (Figure 4) were heated in a chamber furnace to a temperature of 1150 °C.

Then, the heated billets were placed in the gripper of the rolling mill (Figure 5a). After the gripper with the workpiece secured in it took its initial position, the tools were set in rotary motion and, simultaneously, were moved radially in the direction of the axis of the workpiece, penetrating into the material to a depth corresponding to the adopted value of cross-sectional area reduction. While the tools were penetrating into the material, the gripper with the workpiece was set in translational motion and moved along the rolling axis (Figure 5b), which resulted in reduction of the cross-sectional area along the entire length of the rolled material. After the material left the tool working space (Figure 5c), the translational movement of the gripper was stopped, and the rolled rod was removed.

## 4. Experiment Results of Skew Rolling Process of Bimetallic Rods

As an effect of the rotational action of the tools on the moving workpiece, the cross-sectional area of the workpiece was reduced and the two materials were welded together as a result. The semi-finished products obtained during the tests are shown in Figure 6. Due to the specific character of the proposed technology, the rods formed had an allowance (an unformed end), which was used to grip the workpiece. It should be noted, however, that in industrial applications of the technology, the rolling process is designed to run continuously without leaving any allowances. An analysis of the rods obtained in the experiments shows that skew rolling of bimetallic rods in which the outer sleeve is made of X5CrNi18–10 alloy steel results in rather large surface irregularities (helical grooves) (variants I—IV). By contrast, rods rolled from billets in which the outer sleeve is made of S355 steel, have a good surface quality (variant V). This can be explained by the greater plasticity of X5CrNi18–10 steel, which promotes the formation of helical irregularities on the surface. The rolled bimetallic rods were cleaned, and then cross-sections were prepared in order to assess the quality of the joint (weld) between the two materials (Figure 7). The examination of the cross-sections showed that in the case of bimetallic rods rolled in variant IV of the experiment (Table 1), the materials were not joined together (Figure 7d). For the remaining variants, no clear separation of the materials was observed.

The experiments show that there is a strong relationship between process parameters and the quality of the welded parts. It can be seen that billets in which both the core and the outer sleeve are made of unalloyed steels (Figure 7e) are more easily welded together during the skew rolling process than billets in which the outer sleeve is made of high-alloy steel. This can be explained, among others, by large differences in the chemical composition of the two elements of the billet (in the latter case), as well as by their different thermal expansion coefficients (the sleeve made of alloy steel has a much higher thermal expansion coefficient than the core made of non-alloy steel). As a result, during heating in the chamber furnace, contact between the surface of the core and the surface of the sleeve bore was lost. The separation of the two materials caused by differences in thermal expansion led to the formation of oxides on the surface, which made it difficult to weld the sleeve and core together and form a joint between them later on during rolling (Figure 7d). Surface oxides did not form during the rolling of workpieces made of carbon steels with similar thermal expansion coefficients, which meant the materials were easier to weld. An analysis of the cross-sections of the rolled bimetallic rods (Figure 8) shows that not all workpieces were joined along the entire circumference. This is especially prominent in the case of rods made of two different steel grades. In Figure 8b, a clear split can be seen, which demonstrates that the materials were not completely welded along the circumference. This phenomenon was observed for outer sleeves with a low wall thickness and for high the reduction ratio, (defined as *δ* = *D/d*). In experimental variants in which sleeves with thicker walls were used or the reduction ratio was low, the materials were completely welded. Importantly, during the rolling of rods made of carbon steel, a weld was obtained around the entire circumference.

These observations show that the grades of the pair of rolled materials has a large impact on the quality of the joint. It turns out that better welding results are obtained when materials of similar grade are used. Conversely, welding of materials that differ considerably in their chemical composition is much more difficult. In addition, the fact that better welded joints were obtained for billets made of two similar grades of steel can be explained by the greater rigidity of the outer sleeve of those billets (the wall thickness of the sleeve was 8 mm, which constituted 15.7% of the outer diameter of the sleeve. In the remaining cases, this ratio was 5.2% and 8.7%, respectively). In addition, the value of cross sectional area reduction in this case was lower compared to those adopted in the other variants. This observation is quite important because when specimens with the same area reduction were compared, those with thinner sleeve walls were not completely welded. Ineffective fastening (welding) of the two materials rolled in compliance with the parameters of process variant IV may result from too low rigidity of the outside tube compared to the assumed reduction ratio (the ratio of relative tube wall thickness to outside diameter was *D/t* = 0.052, while the reduction ratio in process variant IV was *δ* = 1.285). Consequently, the tube wall was expanded by the considerably more rigid core, which caused strong ovalization of the cross section of the tube as well as loss of contact with the core, which—in turn—made it impossible to ensure permanent welding of both materials. This observation is confirmed by the fact that when bimetallic bars are rolled using the same reduction ratio and semi-finished products with higher tube wall thickness (process variant III − *D/t* = 0.087), complete welding of both materials is obtained. The value of the reduction ratio has an inverse effect on the possibility of producing permanent joints. The results demonstrate that the use of smaller cross-sectional reduction (i.e., lower reduction ratios) facilitates the welding process. As a result, it is possible to ensure permanent joining of both materials with lower reduction ratios, even when the wall thickness of the outside tube is low (process variant II).

The analysis of the microstructure of the weld area (for fully welded materials in which the outer layer was made of high-alloy steel, and the core was made of carbon steel) shows considerable decarburization of the core layer directly in contact with the surface of the sleeve bore (Figure 9). This may be associated with the decarburization of the surface layer of the core which occurred already during the heating process as a result of contact of the material with the atmosphere of the furnace. Later on, after the two component materials had been welded, the surface of the core was further decarburized as an effect of diffusion of carbon into the sleeve. Decarburization intensified as the time and temperature of heating and rolling increased. Decarburization of the surface layers of the core and diffusion of carbon into the sleeve can adversely affect the strength properties of these materials (formation of internal notches). As an effect of a change in the chemical composition of the two materials in the area of the weld, it may also cause intergranular corrosion. The photographic image of the microstructure of the weld clearly shows the boundary between the two materials, which may consist of impurities and residual oxides formed on the surface during heating. Decarburization and oxidation of surface layers can be avoided by changing the heating method (heating in a vacuum or in an inert atmosphere). Another way to reduce the size of the decarburized area is to change the specimen preparation method by preventing air from entering between the heated materials, e.g., by completely closing (e.g., fusion welding) the front surfaces of the materials.

The microstructure of rolled carbon steel bimetallic rods is much less heterogeneous (Figure 10). The core, made of steel with a higher carbon content (42CrMo4), has a predominantly pearlitic structure. The outer layer, made of S355 steel, has a ferritic-pearlitic structure. The welding zone between the two materials (seen as a lighter line in Figure 10) consists mainly of ferrite, which confirms our earlier observation that the near-surface layers of the metal become decarburized when the blanks are heated in a chamber furnace. Even so, the transition zone between the materials is much narrower and its structure is much more uniform compared to bimetallic bars rolled from materials with large differences in the content of carbon and alloying elements.

## 5. FEM Analysis of the Skew Rolling Process of Bimetallic Rods

In the study, the skew rolling process of bimetallic rods was also modeled, mainly to determine what phenomena occurred in outer layer and core materials. FEM numerical analysis of the process was performed using Simufact Forming software (v.15, MSC Software Company, Hamburg, Germany). For the purposes of the calculations, a numerical model of the rolling process was developed in which the kinematic and geometric parameters were identical to the experimental parameters. The geometric model of the process is shown in Figure 11 (variant I in Table 1). The model included three rollers positioned obliquely in relation to the rolling axis, a gripper, and a workpiece. The workpiece was composed of two elements: a core and a sleeve, which were modeled as rigid-plastic objects using eight-node first-order elements. The material models of the workpiece were taken from the material library of Simufact Forming software [29]. For the core, the material model of C45 steel was described by Equation (1):(1)$$\sigma_{p}=1521.306\cdot e^{-0.00269\cdot T}\cdot \varepsilon^{-0.12651}\cdot e^{-0.05957/\varepsilon }\cdot {\dot{\varepsilon }}^{0.14542}.$$

For the outer sleeve, the material model of X5CrNi18–10 steel was described by Equation (2):(2)$$\sigma_{p}=4728.01\cdot e^{-0.00323599\cdot T}\cdot \varepsilon^{\left(-9.05691e^{-6}\cdot T+0.0842687\right)}\cdot e^{\left(\frac{2.4019e^{-5}\cdot T-0.027407}{\varepsilon }\right)}\cdot {\dot{\varepsilon }}^{\left(0.000248575\cdot T-0.154972\right)},$$
where *T*—is the temperature (ranging 700–1250 °C), *ε*—is the effective strain, $\dot{\varepsilon }$—is the strain rate.

All the tools were modeled as perfectly rigid objects. During the process, the rollers rotated in the same direction at a constant speed of 60 rpm and at the same time moved radially towards the axis of the billet at a speed *v* = 5 mm/s (radial movement was stopped when the required cross-sectional area reduction was achieved). The billet was mounted in the gripper, which moved along the rolling axis at a constant speed *v* = 15 mm/s. The contact surface between the individual elements was described using a constant friction model. The friction coefficient was assumed to be: *m* = 0.7 between the billet and the rollers [30,31,32], *m* = 1 between the billet and the handle, and *m* = 1 between the core and the outer sleeve of the billet.

Additionally, in the model of friction between the components of the billet, a full stick contact condition. was introduced (using a window which moved along with the workpiece and was located in the calibrating zone of the rollers) to model the process of welding of the outer sleeve with the core. Of course, this was a fairly large simplification of the phenomena occurring during rolling and welding, however, it seemed acceptable, given the preliminary nature of the present study. The other parameters used in the calculations included: initial temperature of the billet (core and sleeve)—1150 °C and temperature of the tools (rollers and gripper)—100 °C. The heat transfer coefficients between billet material and tools and between core material and sleeve material were 10 and 20 kW/m^2^K, respectively.

The primary objective of the study was to determine the kinematics of material flow and the distribution of stresses in the tool-impact zone, because these parameters have a decisive impact on the feasibility of welding two materials. In skew rolling of bimetallic bars, there occurs high inhomogeneity of strains between the outer material (tube) and the core of the semi-finished product (Figure 12). This normally testifies to increased material flow in the sleeve, which leads to strong thinning of the outer wall. It should be noted, however, that in the present experiment, the increase in strain values was largely due to the occurrence of redundant strain (in the circumferential direction), generated mainly by frictional forces, which did not change the shape of the sleeve, but only increased effective strain in its outer layers. There is also visible strain in the core of the workpiece, which is a favorable phenomenon. What may be alarming here is the large difference in strains between the outer sleeve and the core in the area of direct contact between the two materials (in the calibration zone, the strains in the outer sleeve reach the value of 5, whereas those in the near-surface layer of the core are only 2.8). Such a gradient of strains may lead to the separation of already welded materials.

Material welding is possible due to adhesion and diffusion of materials in the direct contact zone between joined materials [23,33]. For both phenomena to occur, it is indispensable to ensure high temperature and high loads on the contact surface between joined materials. In the analyzed variant of skew rolling, both of these conditions were satisfied: the billets were heated at the beginning of the process to the temperature of 1150 °C; pressure, on the other hand, was the effect of the rotating action of the tools (rollers) on the moving billet. In order to investigate the nature of stresses in the tool–workpiece contact zone, radial (Figure 13), circumferential (hoop) (Figure 14), and reduced stress (Figure 15) distributions were determined during FEM modeling. As expected, the highest compressive stresses were observed in the forming zone (in the area of action of the conical surfaces of the tools). Radial stresses (*σ_R_*) were located directly in the zone of action of the conical roll surfaces, and had values close to zero in the remaining areas. During the process, maximum radial stresses changed their location as the material rotated. Stresses of this type make welding difficult (relatively small areas of high compressive stress neighbor on large areas in which compressive stresses are close to zero).

The distributions of circumferential stress in the tool–workpiece contact zone have a slightly different nature (Figure 14). The stress maps in Figure 14 show compressive stress fields that are much more favorable from the point of view of the welding process. It can be seen that the area of compressive stress covers a large part of the circumference in the contact zone between the sleeve and the core (in the area of action of the conical surfaces of the rollers), which makes it easier to weld the two materials.

Interesting information is also provided by the distribution of Huber–Mises reduced stress (Figure 15). The highest values of reduced stress were observed in the outer sleeve (in the tool–workpiece contact zone). This location was also characterized by a large heterogeneity of stress values, caused by cyclic compression of the rotating material. By contrast, reduced stresses in the core were nearly uniform. Only in small areas (located directly under the tools), were elevated values of reduced stress observed. This character of distribution of reduced stress is not favorable for the welding process and may even cause later separation of materials welded in the initial phase of the process. Because of the simplification used in modeling the process, we did not observe separation of the materials in the FEM model. Further research is required to obtain unambiguous and reliable results. Future studies should be focused on refining the numerical model, among others, by providing the possibility of mapping the stress field in the tool–workpiece contact zone to connections or lack of connections between the nodes of the mesh describing the workpiece.

## 6. Summary and Conclusions

Growing interest in bimetallic materials spurs the search for effective methods of producing semi-finished and finished products from such materials. The present study (especially the experimental part) shows that bimetallic rods can be produced in the process of skew rolling. It seems that the proposed method can be successfully used in the production of bimetallic rod blanks or finished forgings (chiefly stepped axles and shafts), allowing more effective use of construction materials and design of the physical-chemical properties of parts produced from these materials.

The results lead to the following conclusions:The process of skew rolling in a three-roller mill can be used to produce bimetallic rods from materials with different chemical compositions and a wide range of geometric parameters.Welding is much easier at lower cross-sectional area reduction values.The adoption of large cross-sectional area reduction values and the use of outer sleeves with small relative wall thicknesses makes rolling difficult and may make it impossible to weld the two materials.It is much easier to weld materials that have a similar chemical composition.The microstructure of the welding zone shows a large heterogeneity, which is the greater, the larger the difference in the chemical composition of the welded materials.Decarburization of the outer layer of the core was observed directly in the welding zone, which was related to the heating method used.Billets for the rolling process should be heated in a protective atmosphere to reduce the thickness of the decarburized layer and minimize the amount of scale formed on the surface.FEM results point to large differences in strain between the outer sleeve and the core which may have a considerable impact on the welding of the two materials.

## Figures and Tables

**Figure 1 materials-14-00018-f001:**
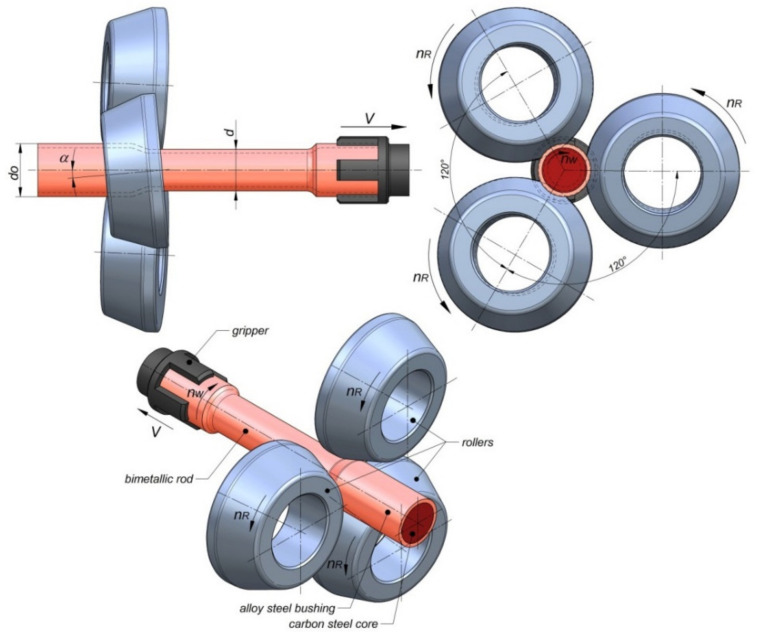
Schematic of skew rolling of bimetallic rods.

**Figure 2 materials-14-00018-f002:**
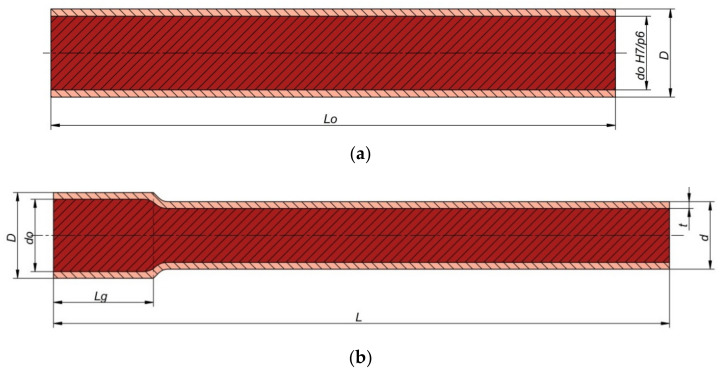
Shape and dimensions of (**a**) billets for rolling bimetallic rods and (**b**) a finished bimetallic rod.

**Figure 3 materials-14-00018-f003:**
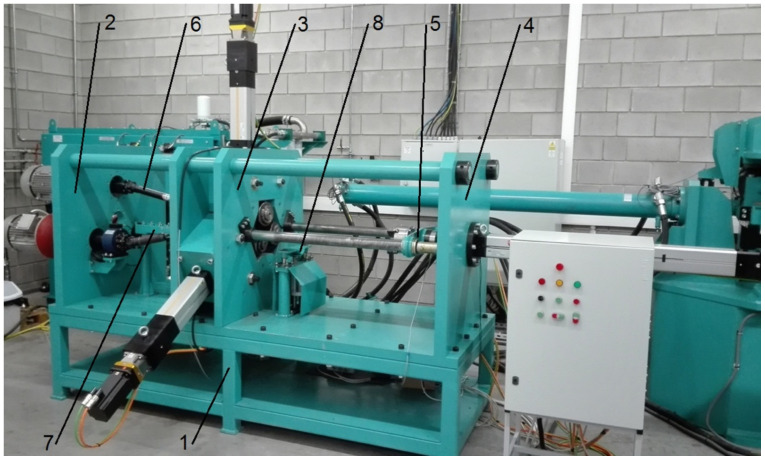
Rolling mill used in bimetallic rod rolling tests: 1—support frame; 2—drive unit; 3—rolling cage; 4—axial travel unit; 5—gripper and workpiece guide units; 6—torque transmission; 7—material feeding unit; 8—workpiece support unit.

**Figure 4 materials-14-00018-f004:**
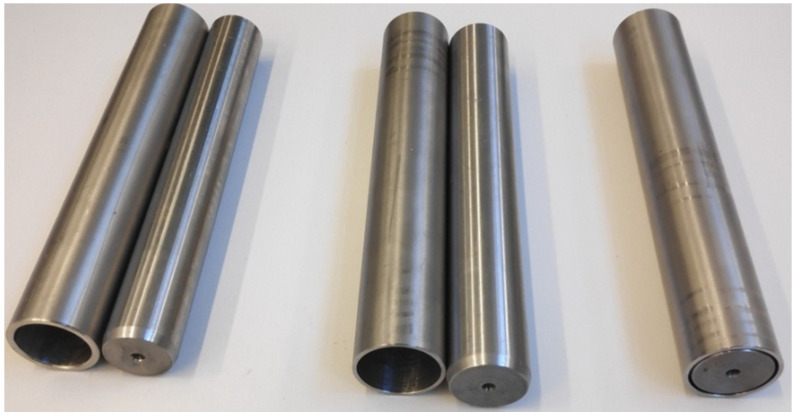
Billets consisting of a sleeve and a core prepared for the rolling process.

**Figure 5 materials-14-00018-f005:**
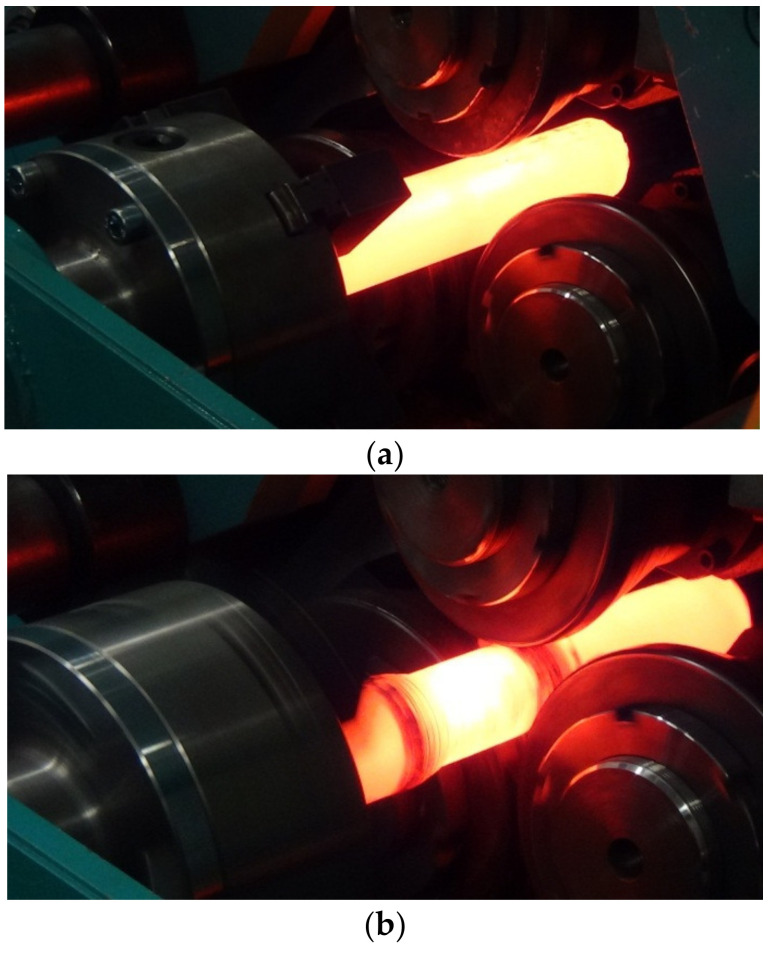
Stages of skew rolling of a bimetallic rod: (**a**) the billet is fed between the rotating tools; (**b**) rolling; (**c**) the rolled part is removed from the tool working space.

**Figure 6 materials-14-00018-f006:**
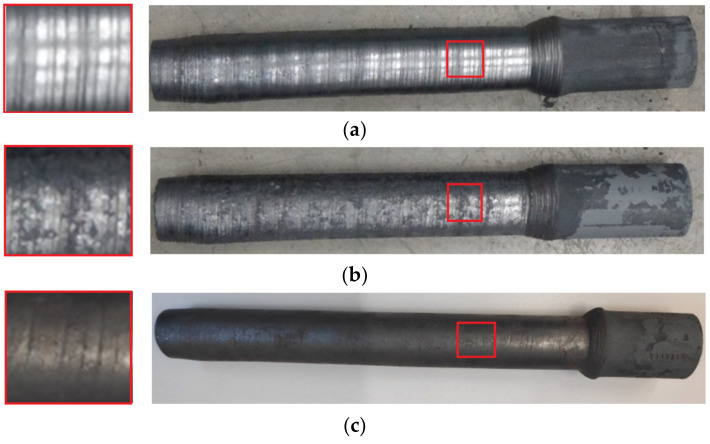
Bimetallic rods rolled during the tests: (**a**) variant III, (**b**) variant IV, (**c**) variant V.

**Figure 7 materials-14-00018-f007:**
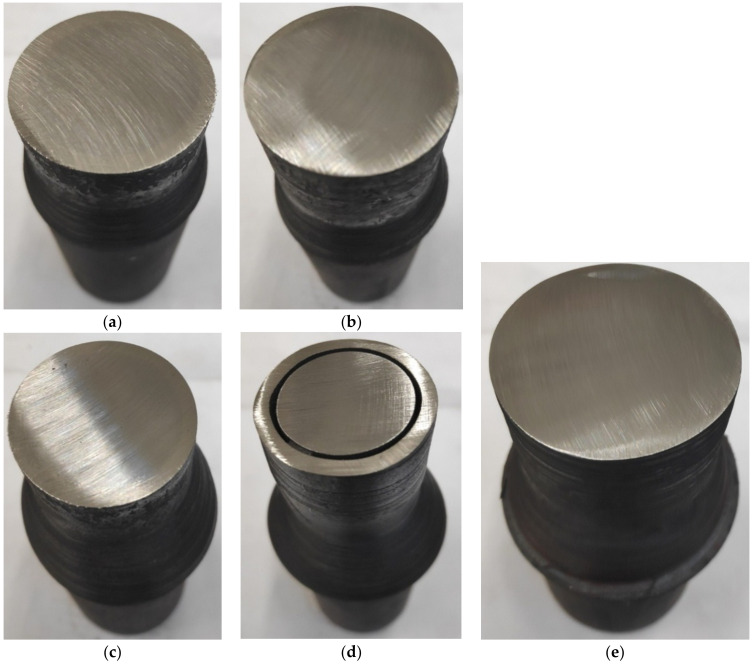
View of cross-sections of bimetallic rods: (**a**) variant I, (**b**) variant II, (**c**) variant III, (**d**) variant IV, (**e**) variant V of the experiment.

**Figure 8 materials-14-00018-f008:**
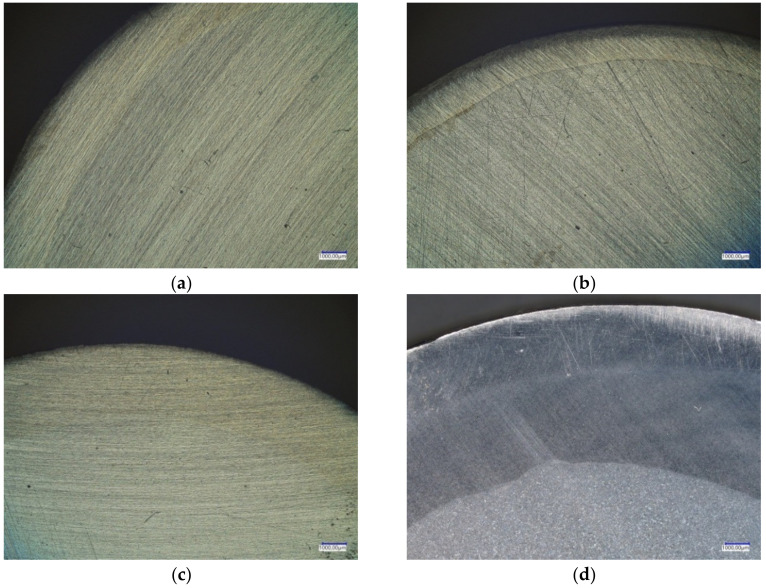
View of the surfaces of cross-sections of bimetallic rods: (**a**) variant I, (**b**) variant II, (**c**) variant III, and (**d**) variant V of the experiment (as described in Table 1).

**Figure 9 materials-14-00018-f009:**
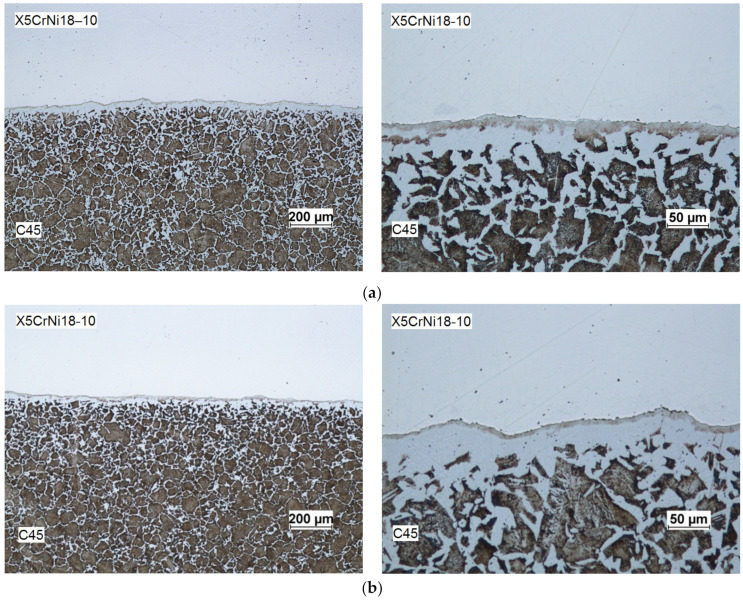
Microstructure of rolled bimetallic rods (a X5CrNi18–10 steel outer layer and a C45 steel core): (**a**) I variant of the research; (**b**) II variant of the research.

**Figure 10 materials-14-00018-f010:**
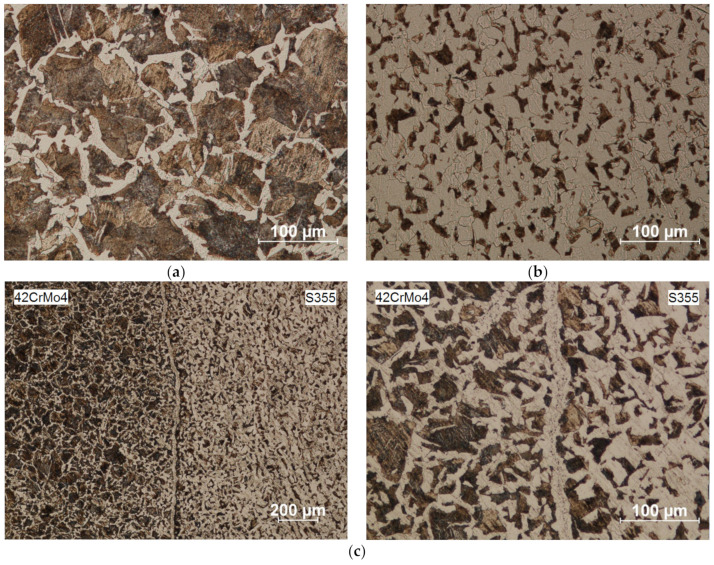
Microstructure of rolled bimetallic rods (**a**) S355 steel outer layer and a 42CrMo4 Scheme 42. CrMo4 steel; (**b**) microstructure outer layer—S355 steel; (**c**) microstructure of the welding zone.

**Figure 11 materials-14-00018-f011:**
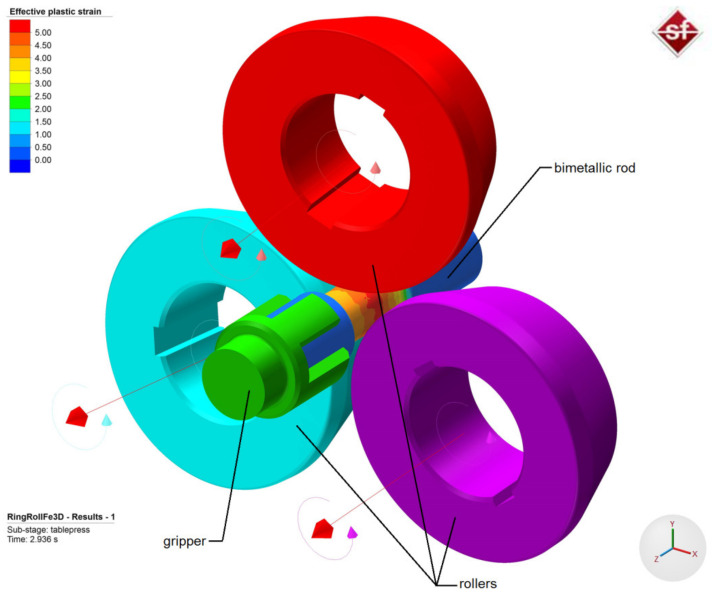
Geometric model of the process of skew rolling of bimetallic rods.

**Figure 12 materials-14-00018-f012:**
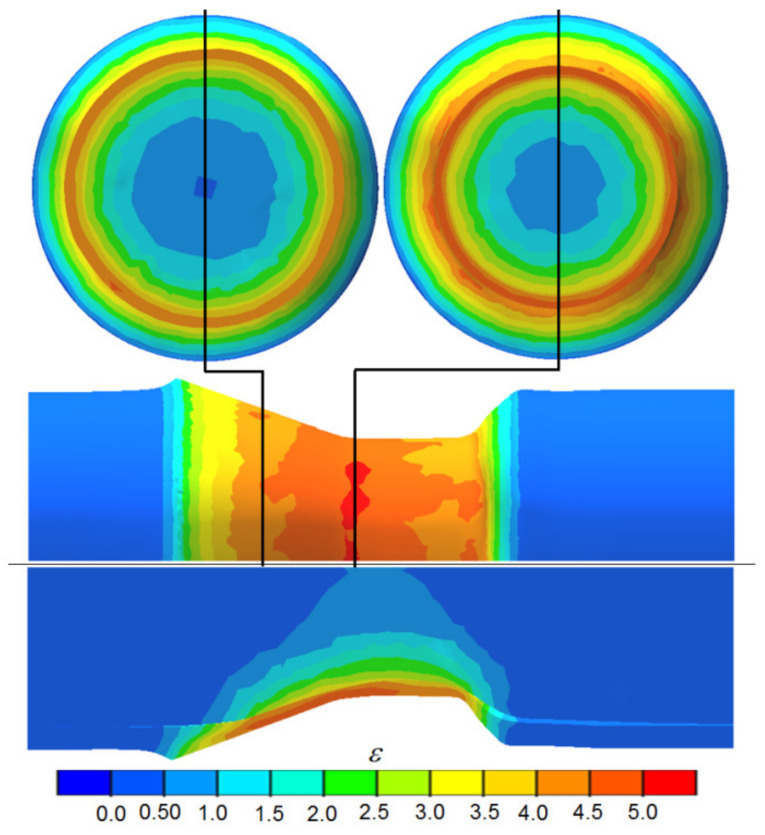
Distribution of effective plastic strain during the rolling of bimetallic rods modeled by FEM.

**Figure 13 materials-14-00018-f013:**
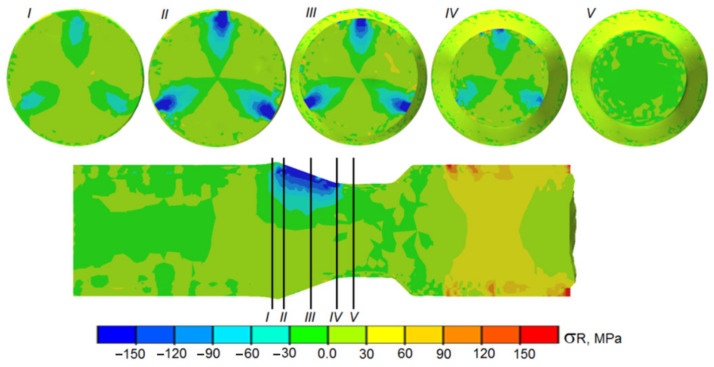
Distribution of radial stress in the tool–workpiece contact zone during the rolling of bimetallic rods modeled by FEM.

**Figure 14 materials-14-00018-f014:**
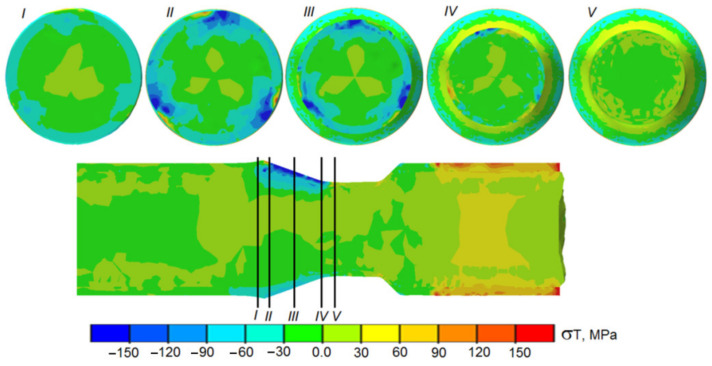
Distribution of circumferential stress in the tool–workpiece contact zone during the rolling of bimetallic rods modeled by FEM.

**Figure 15 materials-14-00018-f015:**
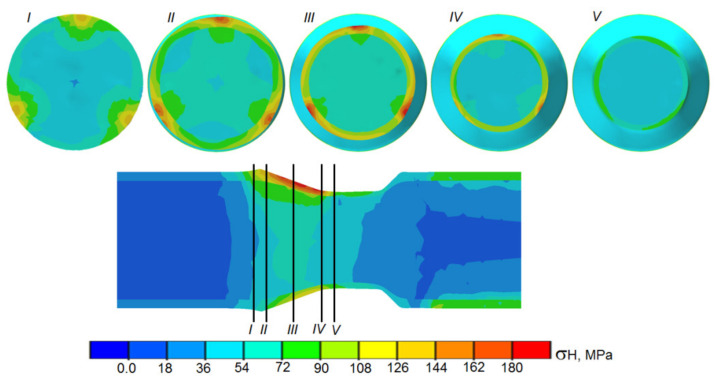
Distribution of reduced stresses in the tool–workpiece contact zone during the rolling of bimetallic rods modeled by FEM.

**Table 1 materials-14-00018-t001:** Geometric parameters of billets and rolled bimetallic rods.

Variant/Parameter	*do,* mm	*D,* mm	*d,* mm	*Lo,* mm	*L,* mm	*Lg,* mm	*t,* mm	*δ = D/d*	Sleeve/Core
I	35	42.4	37	250	300	80	3.7	1146	X5CrNi18–10/C45
II	38	42.4	37	250	300	80	2.2		
III	35	42.4	33	250	360	80	3.7	1285	
IV	38	42.4	33	250	360	80	2.2		
V	35	51	40	250	350	80	8,0	1275	S355/42CrMo4

## Data Availability

Data sharing not applicable.

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
