# Peer review of "Skew Rolling of Bimetallic Rods"

_materials, 2020, doi:10.3390/ma14010018_

Round 1

Reviewer 1 Report

This paper is a preliminary study of the effect of bimetallic rods applicated in skew rolling on the welding process and related properties. It shows a great interest to the reviewer because of pointing out a negative impact on the welding process. However, there are some un-prefer expressions and analysis as following:

(1) In the part “Introduction”, if you want to list an example application of bimetallic materials, you must mark with related literature. And some results which you summarized from reports or studies, you should also mark with literature.

(2) The expression in scientific research is not well using the first person state.

(3) Five variants can be listed in a form that is very visible to reviewers. There only are two variants stated in the second part and the other three variants are not explained in the paper. If the parameters of five variants are listed in Table1, please name it as stated in the paper.

(4) You should better write the name of different parts in Fig 3.

(5) In The analysis of the relationship between wall thickness and good weld joints, you can state which variant has a good thickness ratio, then pointed out which reason leading to this phenomenon.

(6) Figure 8 (d) has bad quality, suggesting you can change a better one.

(7) In figures 9 and 10, there is no alphabet marking the pictures. In Figure 9, there is only one microstructure of one steel. Can you show us the whole microstructure of these bimetallic rods of Fig 9? What are the changing rules of the welding zone in different variants?

(8) You should mark the names of different rollers in Fig 11.

(9) There are some mistakes in the punctuation and you should capitalize the first letter. (Sentence 239)

(10) There is two Fig 11, which one is stated in the analysis of FEM

(11) There is a mistake in the English expression of sentence 254.

(12) If the gradient of strains leads to the separation of variant ⅴ?

Author Response

The authors wish to thank the Reviewer for the in-depth review of their manuscript and all valuable comments and suggestions.

The authors have revised the manuscript in compliance with the Reviewer’s suggestions. Attached are their replies to the questions posed by the Reviewer.

Reviewer 2 Report

The authors study the fabrication method of bimetallic rod using a skew rolling process. Experiment and simulation of the skew rolling process were performed. Following comments are given:

  1. Lines 54-57. The sentence is too long and hard to understand. 
  2. The initial press fit value H7/p6 was used for pre-joining. Why was this value selected? How did the author confirm this value? What is actual compression/interference value between inner core and outer sleeve?
  3. Why was the skew rolling method selected? Did this method provide benefits over traditional method?
  4. Line 84. The authors mentioned the axial component of the tool peripheral velocity. It is hard to understand this velocity.  
  5. The billet was heated up to 1150 oC before rolling. What is the reason behind this temperature selection?
  6. Section 3 should be divided into 2 sub-sections: "Experiment setup" and "Experiment results"
  7. What's the logic (reason) of the experiment plan in Table 1?
  8. Why there is no mechanical join between outer sleeve and inner rod for variant IV (Figure 7d).
  9. Lines 137-139: Except Figure 7d, all other panels in Figure 7 are very similar. Hence, it is hard to conclude that which one having more irregularities on joining surface. 
  10. Line 156. What is the definition for the cold work ratio?
  11. Lines 162-163. This conclusion is trivial. 
  12. Lines 184: "...impurities and residual oxides formed on the surface during" --> There is no figure supporting this conclusion. EDX line scan results through the contact surface should be used.
  13. There was no comparison/relation between simulation (section 4) and experiment (section 3). 
  14. Line 222: m = 0.7, the friction coefficient was chosen randomly without any reason.
  15. Line 297: especilly

Author Response

We wish to thank the Reviewer for the in-depth review of our manuscript and all valuable comments and suggestions. Attached are our explanations to the problems raised by the Reviewer.

Round 2

Reviewer 1 Report

All correction was made.

Reviewer 2 Report

The manuscript was revised. It can be accepted for publication.